# DDNet: Depth Dominant Network for Semantic Segmentation of RGB-D Images

**DOI:** 10.3390/s24216914

**Published:** 2024-10-28

**Authors:** Peizhi Rong

**Affiliations:** Division of Science, Engineering and Health Studies, School of Professional Education and Executive Development, The Hong Kong Polytechnic University, Hong Kong 999077, China; peizhi.rong@speed-polyu.edu.hk

**Keywords:** indoor semantic segmentation, convolutional neural network, RGB-D images, information fusion

## Abstract

Convolutional neural networks (CNNs) have been widely applied to parse indoor scenes and segment objects represented by color images. Nonetheless, the lack of geometric and context information is a problem for most RGB-based methods, with which depth features are only used as an auxiliary module in RGB-D semantic segmentation. In this study, a novel depth dominant network (DDNet) is proposed to fully utilize the rich context information in the depth map. The critical insight is that obvious geometric information from the depth image is more conducive to segmentation than RGB data. Compared with other methods, DDNet is a depth-based network with two branches of CNNs to extract color and depth features. As the core of the encoder network, the depth branch is given a larger fusion weight to extract geometric information, while semantic information and complementary geometric information are provided by the color branch for the depth feature maps. The effectiveness of our proposed depth-based architecture has been demonstrated by comprehensive experimental evaluations and ablation studies on challenging RGB-D semantic segmentation benchmarks, including NYUv2 and a subset of ScanNetv2.

## 1. Introduction

SEMANTIC segmentation is one of the most popular applications in the field of computer vision [1], and it can provide high-definition maps (HD maps) for autonomous mobile robots. Indoor semantic segmentation [2] is a more challenging task for robot mapping and localization because of the uncertainty and complexity of indoor scenes.

Crucial success in semantic segmentation has recently been achieved by convolutional neural networks with the pixel-wise assignment of images to specific categories. A fully convolutional network (FCN) is one of the most prevalent networks with an end-to-end architecture. Numerous state-of-the-art networks have been used with the FCN approach to better parse scenes [3,4,5]. Consequently, encoder–decoder architecture [6,7,8] based on an FCN has been proposed to restore image resolution smoothly, as shown in Figure 1. This architecture is also leveraged in our network.

There are some well-known restrictions related to single RGB images. For example, color cameras fail in the dark and objects in a picture might be blurred due to too much light (some cases are shown in the first column of Figure 2). Therefore, various sensors are exploited as additional channels, such as lidar and depth-sensing cameras, and some even use thermal images to compensate for structural flaws due to color cameras [9]. With the development of depth-sensing cameras (such as Kinect, RealSense, and others) and feasible fusion strategies, depth features have gradually become the mainstream module to assist RGB semantic segmentation. Still, some approaches directly use depth images as the fourth channel of RGB images in the early stage, but this do not work so well because this will cause inconsistencies in network input. Subsequently, some researchers [10,11,12,13] employed a CNN branch, the same as the RGB branch, to extract depth information from the depth map. They then exploited different architectures to fuse the depth feature maps with the RGB branch. Better performances in terms of the segmentation accuracy and visual effects were achieved by these studies than by those using a single branch. Hence, increasingly, scholars have used these strategies to tackle semantic segmentation problems in complex scenes. Meanwhile, their proposed models can be experimentally verified thanks to the open source nature of some of the large-scale RGB-D datasets, including NYUv2 [14], SUNRGBD [15], and ScanNetv2 [16]. 

However, the issues with using single RGB images mentioned above are not completely solved by networks, in which priority is given to the RGB branch. For example, in a state-of-the-art network in which the RGB branch is prioritized, one item can be mistakenly identified as two different objects in a strongly exposed image. This scenario is illustrated in the second row of Figure 2. Thus, when extracting feature information from RGB images using a single CNN branch or using the RGB branch as the main branch, a neural network can easily amplify the drawbacks of blurred boundaries in RGB images. In semantic segmentation, it is crucial to note that segmentation itself is the priority, while assigning semantic information to the segmented region is secondary. Consequently, after clarifying the primary and secondary issues, this paper adopts a dual-branch CNN approach. The depth image is used as the main branch to extract geometric information (such as the shape and contour information), addressing the contour segmentation problem first, while semantic information (such as color and texture features) from the RGB image is used as an auxiliary to ultimately complete the overall semantic segmentation task. We do not directly fuse the RGB-D into a single CNN branch, because merging the two types of images before the feature extraction would confuse features and obscure critical information unique to each type of image.

Unlike these RGB-based methods, DDNet is proposed to fully exploit geometric information in the depth images of RGB-D datasets to understand indoor scenes. RGB features and depth features were, respectively, extracted by two CNN branches, as shown in Figure 3. Specifically, the author considered that depth images have more precise and stable appearance information. Such images are more suitable for segmentation as primary inputs into an encoder, while RGB images that have blurry, marginal information due to various lighting scenarios could provide semantic information for the depth branch as an auxiliary module. Consequently, to further study which has a more significant influence on the performance of scene parsing, two proportionality coefficients for the fusion weights of RGB and depth feature maps are introduced in this study.

The main contributions of the paper are summarized as follows:A DDNet is proposed in which the depth CNN branch acts as the primary encoder to extract geometric information and the color branch as the auxiliary encoder to extract semantic and complementary appearance information;An effective fusion strategy is proposed in which RGB features are guided to combine with the depth branch at the most reasonable fusion weight.

The best RGB and depth branch fusion weights were found. The network’s results displayed better performances for two convincing criteria (mean accuracy and mean intersection-over-union) than the baseline network and some state-of-the-art methods. All experiments were evaluated on the following two public datasets: NYUv2 and ScanNetv2.

## 2. Related Work

Semantic segmentation is a pixel-wise classification task for images. In this topic, prior works were reviewed. As is well known, fully convolutional networks [3] are the first deep neural networks to make per-pixel predictions for image segmentation. They replaced fully connected layers in the classification net with convolution layers to achieve pixel-wise image segmentation. However, its upsampling operations directly restore the resolution to that of the input image after the last subsampling layer, which is so rough that it discards many extracted objects’ details in the feature maps. Aimed at this issue, SegNet [6] proposed an encoder–decoder architecture in which the decoder has five unpooling layers to gradually restore pixels. Such smooth upsampling operations can preserve features well. RTFNet [9] utilized the same encoder–decoder architecture for the RGB-thermal semantic segmentation of urban scenes. Wang et al. [17] employed convolution operations as their subsampling layers, with deconvolution operations as counterparts; thus, another symmetric encoder–decoder modality took shape. Similarly, an SDN [18] also utilized deconvolutional networks to restore the localization information of feature maps. In addition, it employed multiple encoder–decoder architectures, each of which was a duplicate and regarded as a modular unit. With Refinenet [4] and SCN [19], a top-down cascaded network was applied to incorporate coarse semantic information from deep layers with fine-grained appearance features from shallow layers. In addition, an increasing number of methods [19,20,21,22] are committed to computing contextual information on the encoder part. Some works [23,24,25] balanced cost and performance by reducing the number of parameters, as well as by introducing contextual modules, which dramatically improved real-time operations without compromising accuracy too much.

Contextual information (relationships among adjacent pixels) can improve the performance of semantic segmentation. Previous works [26,27] introduced modified long short-term memory (LSTM) to fuse the spatial interaction between short-distance and long-distance, with feature maps serving as modules of the contextual information. Using PSPNet [28] and PSMNet [29], pyramid pooling modules (PPM) were presented that extract contextual features from several different pyramid scales by global average pooling operations. Then, the multiscale features were merged with the last feature map of the backbone as the final feature. UPerNet [21] is a multitask framework that parses different concepts at various semantic levels, such as parts, materials, objects, textures, and scenes, at the same time, and PPM is directly appended to the scene identification section at the end of the encoder network to achieve remarkable performance. Chen et al. [8] proposed an atrous convolution that has larger receptive fields without increasing the number of parameters or computations. As is well known, the larger the receptive fields, the more comprehensive the multiscale information that is captured (the size of receptive fields can be adjusted by changing the dilation rate). They further developed atrous spatial pyramid pooling (ASPP) based on atrous convolution to capture objects on different scales. Many prior works [20,30] adopted modified ASPP as an encoding module to obtain the semantic context of complex scenes. PSANet [31] predicted each pixel by collecting contextual information from all pixels for a comprehensive understanding. CCNet [32] helped in the prediction by gathering context from adjacent pixels in the horizontal and vertical directions of a sparse attention map to take performance, as well as efficiency, into account. Some methods [19,22,33] segment super-pixels first, which are small areas that have pixels with similar features, such as color, brightness, and texture. They can transform a pixel-level image into a district-level image without destroying the boundary information of objects in the image. Other recent works [8,31,34,35] use conditional random field (CRF) or Markov random field (MRF) as post-processing algorithms to enrich the contextual information and capture fine edge details.

Depth images have much less semantic information than RGB images, but they contain rich geometric and contextual information, like ready-made images with super-pixels, which assists in segmenting objects and scenes. Some prior works [36,37] fed depth data into CNNs, including the horizontal disparity, height above ground, and angle of a pixel’s local surface normal images (HHA). They intended to extract as much information from depth images as from gray-scale images. Fusenet [11] used two branches of CNNs to encode colors and depth images, respectively, and they fused depth information into the feature maps of the RGB branch as a complementary module. Blum et al. [38] applied two statistical fusion approaches (Bayes categorical fusion and Dirichlet fusion) to improve robustness and adaptability after extracting features from RGB and depth images. An SCN [19] used depth information as a cue to guide the aggregation of features in multiple super-pixels by identifying different objects’ co-existences. According to a comparison of the average depths of two adjacent super-pixels, compression or expansion processing was performed. Compression operations can refine features, while expansion can enrich features in regions. Wang et al. [39] took only depth information as the weight of the RGB convolution kernel. They believe that adjacent pixels with the same depth should have stronger correlations. Cai et al. [40] employed depth data and RGB semantic segmentation as inputs to achieve instance completion, as well as 3D scene reconstruction. There are some works [41,42] that used only RGB images for depth estimation, and the other branch of a CNN was used for semantic segmentation; these two tasks can complement and enhance each other.

It is evident that depth information serves as an auxiliary module to the color branch in most 2D semantic segmentation methods. In this way, the semantic information is excessively rich and important geometric information is scarce. In this paper, a depth-based architecture is proposed to extract the features and contexts of objects. Furthermore, depth information plays a more important role by giving it a bigger weight.

## 3. Depth Dominant Network

There are some representative failure cases with the use of Fusenet [11] methods (our baseline network) for semantic segmentation. These cases inspired the proposal of our depth dominant network (DDNet), which is illustrated in Figure 3, to solve the problems related to RGBD datasets. In addition to this network, fusion weights between RGB and depth images were employed to improve the performance.

### 3.1. Representative Failure Cases of RGB-Based Methods

Indoor scenes are complex, and a mass of objects is always irregular in shape (e.g., clothes, chairs, and shelves). Observing the prediction results of the Fusenet network, three types of common issues were collected for scene labeling.

*Shape Accuracy*: Integrity and accuracy should be promised when splitting an item from an image. But because of highlight exposure or dim light, objects that need to be segmented are easily confused with nearby objects of similar color, for example, there will be shadows in a dark room, and a little black chair on a grey floor may lose some geometric information. In the first row of examples in Figure 2, Fusenet confused the trash can in the yellow box with the photo frame because they are close and have similar colors in shade. This led to an extra part being added to the trash can’s structure, resulting in an inaccurate representation of its real appearance. This is a common occurrence when depth is not fully utilized.

*Contextual Relationship*: It is a common phenomenon that a multilayered object is identified as a variety of items by the baseline network. For example, with RGB-based networks, this error is more likely to occur because a single object is always multicolored or displays different brightness levels under uneven illumination. In the second row of Figure 2, Fusenet predicted the object in the yellow box as being part of one item and part of another item. This performance is connected to geometric integrity. In general, it tends to be identified as one object when it looks like a single item or has similar depths in the depth map.

*Missing Identification*: The network occasionally neglects small objects when it overlaps other large labels. Several small objects, like pillows, may be hard to parse on a bed; there is a high probability that it will identify it as the sheet. As shown in the third row of Figure 2, Fusenet failed to recognize the clothes in the box. These small-sized things are evident in the depth images. However, the ground truth even neglected those labels, as shown in Figure 2. These errors in the dataset itself reduce the pixel accuracy of the true prediction results.

Summarizing the above failure cases, those errors should have been solved by the Fusenet network because it employed depth images, but it did not take full advantage of the depth information. Thus, a depth dominant network with fusion weights can reduce those errors in scene parsing.

### 3.2. Network Architecture

The architecture of our proposed Depth Dominant Network (DDNet) is illustrated in Figure 3, featuring a dual-branch design for extracting depth and RGB features, along with a symmetric decoder to restore the feature maps to their original input size. To enhance the contextual relationships between local and global features, we incorporated a pyramid pooling module (PPM) before the symmetric decoder stage.

(1)Encoder Design

The encoder network in our Depth Dominant Network (DDNet) is designed with a dual-branch architecture, in which each branch is dedicated to processing different types of inputs, as follows: one for depth information and the other for RGB images. This design leverages the complementary nature of both data types, enhancing the overall performance of tasks such as semantic segmentation.

Backbone Architecture: For the backbones of both branches, we primarily used VGG-16 [43], known for its effectiveness in capturing spatial hierarchies. The VGG-16 architecture consists of 13 convolutional layers, which are organized into five blocks. Each block contains two or three 3 × 3 convolutional layers, each followed by ReLU activation, and then a 2 × 2 max-pooling layer to downsample the feature maps. Specifically, the first two blocks have two convolutional layers, while the last three blocks have three convolutional layers.

Feature Fusion: To integrate the features from the RGB and depth branches, we performed an element-wise summation of the RGB features into the corresponding depth features before every max-pooling layer. This fusion mechanism allows for the geometric and appearance information from the depth branch to be enriched with semantic details from the RGB branch, thereby improving the precision of object boundary delineation.

Dropout Layers: To prevent overfitting, dropout layers are introduced after the third, fourth, and fifth pooling layers. These dropout layers are applied to the fused feature maps, ensuring that the model generalizes well to unseen data.

Alternative Backbones: In addition to VGG-16, we also experimented with using ResNet50 [44] as an alternative backbone for either the depth or RGB branch. ResNet50 introduces residual connections, which can help mitigate the vanishing gradient problem and allow for deeper network architectures. When using ResNet50, the structure remains similar, with feature fusion occurring at the end of each residual block and dropout layers placed strategically to maintain robustness.

By combining the strengths of the depth and RGB information through this carefully designed encoder, DDNet is able to capture a rich set of features that are both geometrically and semantically informative, leading to improved segmentation results.

(2)Pyramid Pooling Module (PPM)

The pyramid pooling module is designed to enhance the model’s capability in capturing global context, which is essential for understanding object scales and semantics within scenes. The PPM operates by aggregating multiscale contextual information through a series of pooling operations on different scales.

Multiscale Pooling: The PPM employs four distinct pyramid levels with bin sizes of 1 × 1, 2 × 2, 3 × 3, and 6 × 6. At each level, global average pooling is applied to generate pooled representations that cover the whole image, half of the image, and smaller portions, respectively. This allows the module to capture both global and local contextual information.

1 × 1 Bin Size: captures the entire image, providing a global context.

2 × 2 Bin Size: divides the image into four equal parts, capturing a broader but still relatively global context.

3 × 3 Bin Size: divides the image into nine equal parts, capturing a more detailed local context.

6 × 6 Bin Size: further divides the image into 36 equal parts, capturing a very fine-grained local context.

Dimensionality Reduction: After pooling, the dimensionality of the context representation from each level is reduced using a 1 × 1 convolution layer. This step ensures that the weight of the global feature is maintained while reducing the number of channels, making the feature maps more manageable and efficient.

Upsampling and Fusion: The low-dimensional feature maps are then upsampled back to the size of the original feature map via bilinear interpolation. This process allows for the seamless integration of these multiscale features with the initial feature representation. By combining both local and global contextual information, the PPM produces a richer and more comprehensive final feature map that can be used for subsequent tasks, such as pixel-level prediction.

This structured approach ensures that the PPM effectively captures a wide range of contextual information, leading to improved feature representation.

(3)Decoder Design

The decoder part of DDNet was designed to be symmetric to the encoder, excluding the additional layers used for feature fusion. It consists of a series of convolutional, batch normalization, ReLU, and dropout layers, mirroring the structure and operations found in the encoder. While the encoder uses max-pooling to downsample the feature maps, the decoder employs unpooling (or deconvolution when using ResNet50 as the encoder) to upsample them, utilizing the indices from the corresponding max-pooling operations to maintain spatial consistency. This symmetry ensures that the spatial information lost during the downsampling process is recovered, enabling the generation of high-resolution segmentation maps that match the input image dimensions. By combining the strengths of depth and RGB information, along with the incorporation of a PPM, our DDNet achieves state-of-the-art performances on various indoor RGB-D datasets, demonstrating its effectiveness in capturing both local and global contextual relationships for improved scene parsing and semantic segmentation.

### 3.3. Fusion Strategies

As described in Section 2, there is no doubt that using depth as the fourth channel of RGB input directly leads to worse performance. The following four fusion methods [45] were studied and compared: element-wise summation, element-wise multiplication, concatenation, and channel concatenation. Element-wise summation was eventually used because of its stability and effectiveness. The fusion strategy is shown in the red box in Figure 3. Two branches extract RGB and depth information separately. The depth branch are fused with the aligned RGB feature maps before every subsampling operation. So, there are five fusions to fully fuse the semantic information from RGB images with the morphology information from depth feature maps.

To further the research, either the depth image or RGB image had a greater influence on the performance of the scene parsing, and to make the most of the depth information, this paper introduced the following two weights: *α* and *β*. They influence the fusion of depth and RGB, as shown in the following formula:
(1)$$F=\alpha f_{RGB}+\beta f_{depth}$$
where F,
$f_{RGB}$, and $f_{depth}$ denote fused feature map, RGB feature map, and depth feature map, respectively. In addition, *α* denotes the fused weight of the RGB feature input, while *β* denotes the fused weight of the depth feature input. Specifically, the weight of the major semantic information is denoted by *α*, while *β* represents the proportion of geometric information. Although color images also contain morphological information, and they can sometimes appear fuzzy to the network due to uneven lighting or other factors. These fuzzy shapes can lead to inaccurate segmentation. Therefore, color images should not be used as the primary input. We hope that each branch performs its own functions, with color images serving as auxiliary inputs to provide semantic and complementary marginal information.

Accordingly, we propose a weight fusion method, where the main depth image branch is given a larger fusion weight, while the auxiliary module is assigned a smaller fusion weight. This paper compared the criteria to find the optimal combination by adjusting the values of *α* and *β*. Using the single-variable control method, when the fusion weight of the depth image, *β*, was fixed at 1; the fusion weight of the RGB image, *α*, was adjusted in equal gradients with a step size of 0.1 within the range of 0.1 to 1. Similarly, when adjusting the value of *β*, *α* was kept constant. Under these settings, 19 groups of experiments were intensively arranged to find the optimal solution. The idea mentioned above can be confirmed if the best performance occurs when *β* ≥ *α*. The experimental results of this study show that when *α* is set to 0.5 and *β* to 1, the three evaluation metrics achieve optimal performance.

## 4. Experiments

### 4.1. Datasets and Evaluation Metrics

Datasets: Two publicly challenging datasets, NYUv2 [14] and ScanNetv2 [16], were utilized to evaluate the Depth Dominant Network on RGB-D semantic segmentation tasks. Both of them consist of captured indoor scenes.

NYUv2 is a prevailing dataset for indoor semantic segmentation. It includes 1449 RGB-D images with a 320 × 240 resolution, of which 795 images were used for training and 654 for testing. The range of depth values was between 0 and 255. Each pixel in the images was labeled by 40-class settings.

Because of the large volume of the whole ScanNetv2 dataset, the author used a smaller subset: scannet_frames_25k. It includes 25,000 frames which are resampled from the integrated dataset of about 2.5 million frames. This subset also contains 1513 scenes, of which 1201 scenes were used for training, and 312 scenes in the test set called scannet_frames_test were used for testing. It has 21 object labels, which are all contained in nyu40 labels. Color images were provided in the 8-bit “JPG” format and depth images in the 16-bit “PNG” format.

*Evaluation Metrics*: The following three metrics were utilized for comprehensive evaluation: global accuracy (referred to as Global acc), mean intersection-over-union (referred to as Mean IoU), and mean accuracy (referred to as Mean acc). Since different classes of objects appear at different frequencies, low-frequency categories trained will have poorer results. This situation will further lead to a reduced reference for Global acc [11]. Hence, two other criteria were evaluated intensively.

### 4.2. Implementation Details

For most of the experiments, the baseline framework with the pretrained VGG16 was employed. In the ablation study, ResNet50 replaced VGG16 as the depth or color encoder to extract features. To ensure the consistency of the encoder structure, ResNet50 was reproduced and trained from scratch, the details of which are provided in Section 4.1.

Our DDNet was trained using Pytorch. Almost all
operators were implemented with CUDA acceleration on a single GeForce RTX
2080Ti GPU. SGD was chosen as the optimizer with a momentum of 0.9, and the
initial learning rate of 0.005 was multiplied by ${\left(1-iter/max\_iter\right)}^{0.9}$ every 25 epochs. The
batch size was set to 8; and *α* and *β* were set to 0.5 and 1, respectively (details on the selection of *α* and *β* are provided in
Section 4.1).

### 4.3. Experiment Results on NYUV2 Dataset

The results of the comparison between DDNet and other state-of-the-art networks are shown in Table 1. It can be easily observed that the best results for our network outperformed the mean accuracy of the other methods. It achieved 45.7%, which exceeds by 2.3% that of the baseline network. Our network also exceeded the baseline in the Mean IoU. The baseline network was more competitive than ours in global accuracy. However, it is worth noting that DDNet-Weight *α* did not introduce any modules, such as the atrous spatial pyramid pooling module or transformers, which can improve performance. If it employed such modules, DDNet may achieve better results. However, the purpose of this experiment was to validate DDNet’s better performance compared to many state-of-the-art networks by focusing only on geometric information in depth images without introducing prominent modules.

*Ablation Study*: Table 2 reports the influence of different fusion weights on the NYUv2 dataset. It was made sure that only one weight was changed at a time and all reasonable matches were tested as much as possible. The purpose of this experiment was to explore the best match rather than the best performance, so the epochs were set to 100. Upon determining the best match, we trained the model for 250 epochs to achieve the best performance on the NYUv2 test set. When α = 0.5 and β = 1, all three criteria achieved the best results, and only a weight of 0.5 for the color information was needed to provide auxiliary semantic information to the main depth branch. Our idea that depth maps are more important for semantic segmentation was verified.

The other ablation study on NYUv2 was conducted to seek better backbones. VGG16 and ResNet50 were mainly employed. It is well known that the subsampling layer of VGG16 conducts max-pooling operations while ResNet50 relies on stride convolution operations in their four layers. To ensure the symmetry of the encoder–decoder architecture and the consistency of the different backbones, VGG16 and ResNet50 were reproduced and trained from scratch. In addition, we resolved the four layers of ResNet50 and replaced stride convolution with max-pooling so that the corresponding unpooling on the decoder could use the above pooling indices. The RGB branch and depth branch employ permutation and a combination of VGG16 and ResNet50 to determine their own best backbones. As is shown in Table 3, the depth branch using VGG16 and the RGB branch utilizing ResNet50 achieved the best performance compared with other permutations and combinations. Interestingly, this experiment was conducted without employing a fusion weight; however, no matter which backbone was used for the RGB branch, as long as VGG16 was used for the depth branch, the three criteria achieved higher values. Here, again, our idea that depth maps are more important in semantic segmentation was verified. Furthermore, the pyramid pooling module was employed at the end of the depth branch to extract more contextual information. It is evident that the addition of this module resulted in significant increases in Mean IoU and Mean accuracy. Figure 4 shows the qualitative comparison results on the NYUv2 dataset.

### 4.4. Experiment Results on ScanNetv2 Dataset

A comparison of the mean accuracies on the ScanNet benchmark between DDNet and its baseline is shown in Figure 5. The networks were trained with a pretrained VGG16. The histogram illustrates that the Depth Dominant Network exhibited a better performance than the RGB dominant network (baseline), and DDNet with fusion weights could further improve the Mean accuracy.

Several examples comparing the baseline method on the ScanNetv2 validation set are shown in Figure 6. Visually, DDNet also performed well on contextual relationships compared to the baseline method. In these examples, the baseline model predicted the object as part of one item and part of another item, such as the “shelf”, “fire extinguisher”, “table”, “trash can”, and “cabinet” in the five rows. DDNet corrects the above errors because the items have obvious contextual relationships in the depth images.

After training the best model on the ScanNet dataset, the per class IoU on the test set was calculated on the official website of the ScanNet benchmark challenge. The results are reported in Table 4. The results of the other state-of-the-art networks were all cited from the ScanNet benchmark leaderboard. These networks include PSPNet and the baseline method. DDNet uses parts of these two methods for reference, but it had a higher Mean IoU than they did. Specifically, DDNet achieved the highest value for IoU in many object categories, a lot higher than the other methods. For example, the greatest improvement was achieved in the bathtub category, with a value of 28.5%, and the second, 26.0%, was evaluated in the shower curtain category, which has many diverse appearances in this dataset, while it exhibited comparable performances to the best results in other classes. Further analyzing these results, many objects were confused by the RGB-based networks. However, one item in the depth images was more consistently confused. Hence, a network dominated by depth information can do a better job in segmentation.

## 5. Conclusions

A depth dominant network for RGB-D semantic segmentation that can extract in-depth geometric information from depth images was proposed in this paper. Two branches of an encoder were utilized to extract color and depth feature maps, respectively. The depth branch served as the primary part of the encoder, while auxiliary semantic information and a few complementary appearance features were provided by the RGB branch. Subsequently, fusion weights for the two branches were introduced to fully leverage the geometric information in the depth maps. According to the results of experiments conducted on the NYUv2 and ScanNetv2 datasets, DDNet’s performance in scene parsing was effectively improved, surpassing the baseline model without the introduction of other existing modules. Our novel and effective fusion ideas—that clear geometric information in depth images is more beneficial for segmentation and that color images may have blurry marginal information because of light intensity or other causes but can still guide semantic learning—were validated by these results. One direction for future work is to adopt the proposed depth-based network for datasets related to fire scenarios. As RGB images become more ineffective, depth images will be treated as perfect feature extraction sources in fire scenes.

## Figures and Tables

**Figure 1 sensors-24-06914-f001:**
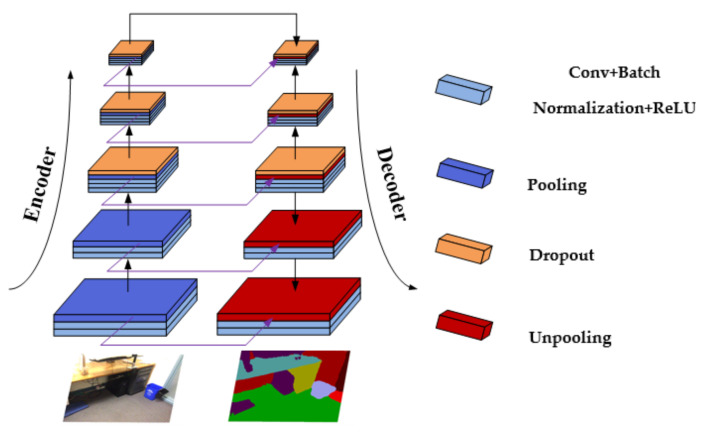
An illustration of the simple symmetric encoder–decoder network.

**Figure 2 sensors-24-06914-f002:**
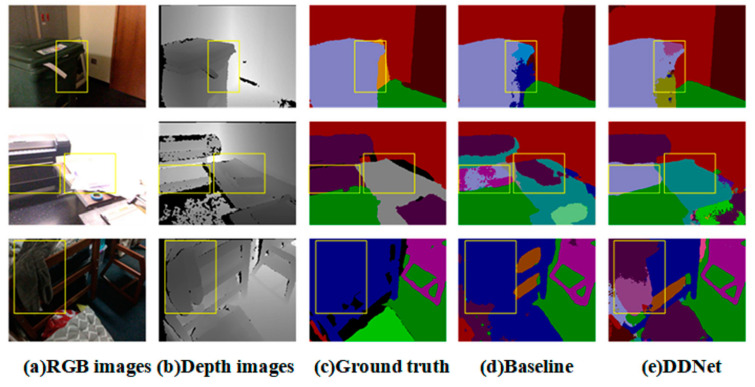
Some representative failure cases on the ScanNetv2 [16] dataset. The first row shows that the baseline model lost morphological integrity and accuracy in shade, leading to an extra part of the trash can’s structure. The second row shows that Fusenet confused part of a single item as other objects while adjacent pixels of the same object have evident contextual relationships in the depth image. The third row shows that both the prediction by the baseline model and the ground truth neglected the clothes on the bed.

**Figure 3 sensors-24-06914-f003:**
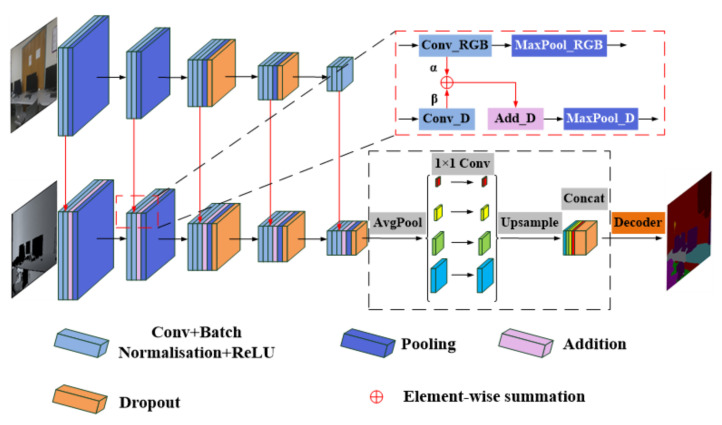
An illustration of the proposed DDNet. There are two branches of CNNs, which are used to extract RGB and depth features, respectively. The depth branch is the main part of the encoder, while the RGB branch is the auxiliary encoder, because the depth image has more direct geometric and appearance information which is beneficial to segmentation. The fusion details in the red box show that the RGB feature map merges with the depth feature map through element-wise summation with two fusion weights. In addition, the pyramid pooling module is employed before the symmetric decoder to strengthen the contextual relationship.

**Figure 4 sensors-24-06914-f004:**
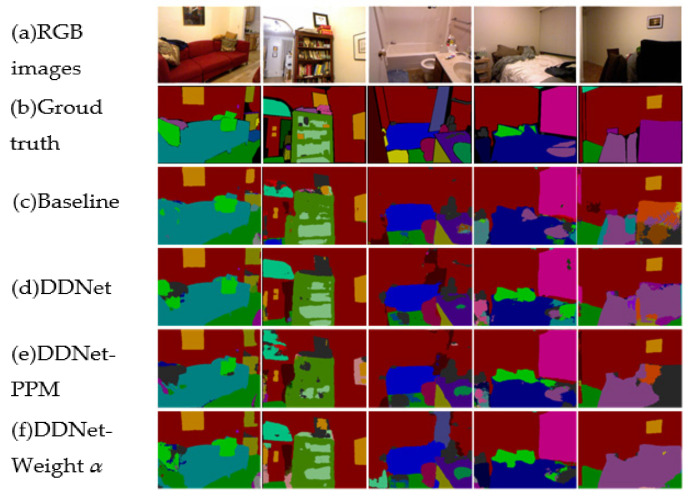
Visual improvements in the NYUv2 test set. “DDNet-PPM” denotes depth dominant network with the pyramid pooling module, and “DDNet-Weight *α*” denotes depth dominant network with an RGB fusion weight of 0.5 and a depth fusion weight of 1.

**Figure 5 sensors-24-06914-f005:**
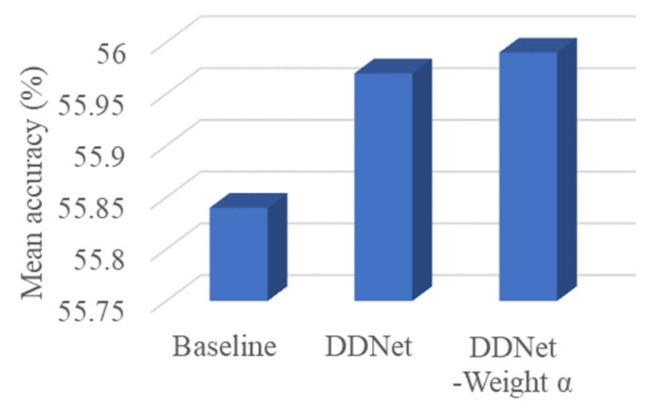
Increases in the mean accuracies with the addition of the main contributions.

**Figure 6 sensors-24-06914-f006:**
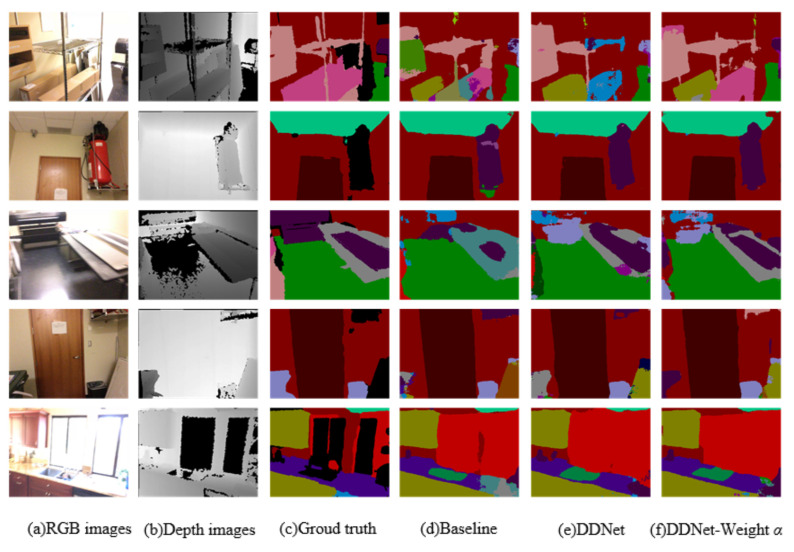
Visual improvements compared with baseline method on ScanNetv2 validation set. Our “DDNet-Weight α” network yields accurate and best results.

**Table 1 sensors-24-06914-t001:** Segmentation results for the Nyuv2 test set in comparison to the baseline network and other state-of-the-art networks. The DDNet was trained with the pretrained VGG16 and the epochs were set to 250 for the best performance. DDNet-Weight *A* outperformed other networks in mean accuracy, and it was also superior to the baseline model in the mean intersection-over-union.

*Methods*	*Global acc (%)*	*Mean IoU (%)*	*Mean acc (%)*
FCN [3]	61.5	30.5	42.4
D-CNN [39]	60.3	27.8	39.3
AdaShare [46]	61.3	29.6	-
EDNAS [47]	58.1	22.1	-
Baseline [11]	66.0	32.7	43.4
DDN	65.2	32.3	44.2
DDN-Weight α	65.1	32.9	45.7

**Table 2 sensors-24-06914-t002:** Performances when using various *A* and *Β* values on the Nyuv2 test set. The networks were trained from scratch, and the epochs were set to 100.

*α*	*β*	*Global acc (%)*	*Mean IoU (%)*	*Mean acc (%)*
0.1	1	57.1	23.9	35.0
0.2	1	59.6	25.8	37.4
0.3	1	59.0	25.7	37.6
0.4	1	59.9	26.1	37.2
0.5	1	**60.0**	**26.9**	**39.3**
0.6	1	59.3	25.1	37.0
0.7	1	57.8	25.4	37.4
0.8	1	59.6	26.0	38.3
0.9	1	59.6	25.6	37.0
1	0.1	58.5	24.5	34.0
1	0.2	58.0	24.4	35.4
1	0.3	57.8	24.2	35.4
1	0.4	57.1	23.4	34.4
1	0.5	57.7	24.3	35.3
1	0.6	57.8	22.7	32.7
1	0.7	58.6	24.5	35.8
1	0.8	59.8	26.0	38.0
1	0.9	58.9	25.8	37.8
1	1	58.7	25.5	37.0

**Table 3 sensors-24-06914-t003:** Performances when using different backbone networks on the Nyuv2 test set. VGG16 and Resnet50 were reproduced and trained from scratch to ensure the consistency of the encoder structure. The pyramid pooling module is denoted as PPM, which was added before the decoder. The epochs were set to 250.

*Models*	*Global acc (%)*	*Mean IoU (%)*	*Mean acc (%)*
Depth (VGG16)-RGB (VGG16)	61.1	25.7	35.1
Depth (ResNet50)-RGB (VGG16)	59.7	24.4	33.2
Depth (VGG16)-RGB (ResNet50)	**61.7**	26.1	35.7
Depth (ResNet50)-RGB (ResNet50)	60.6	25.1	34.3
Depth (VGG16)-RGB(VGG16)-PPM	**61.7**	**29.8**	**41.6**

**Table 4 sensors-24-06914-t004:** Per-class IoU on the ScanNet Benchmark. The results for DDNet and other state-of-the-art methods were evaluated on the official website of the ScanNet Benchmark Challenge.

Method	PSPNet [28]	Mseg [48]	3DMV [49]	AdapNet++ [50]	FuseNet [11]	DDNet
bathtub	0.490	0.505	0.481	0.613	0.570	**0.898**
bed	0.581	0.709	0.612	0.722	0.681	0.660
bookshelf	0.289	0.092	0.579	0.418	0.182	0.185
cabinet	0.507	0.427	0.456	0.358	0.512	0.431
chair	0.067	0.241	0.343	0.337	0.290	0.304
counter	0.379	0.411	0.384	0.370	0.431	0.256
curtain	0.610	0.654	0.623	0.479	0.659	**0.659**
desk	0.417	0.385	0.525	0.443	0.504	0.461
door	0.435	0.457	0.381	0.368	0.495	0.480
floor	0.822	0.861	0.845	0.907	0.903	0.904
other furniture	0.278	0.053	0.254	0.207	0.308	0.293
picture	0.267	0.279	0.264	0.213	0.428	**0.572**
refrigerator	0.503	0.503	0.557	0.464	0.523	0.484
shower curtain	0.228	0.481	0.182	0.525	0.365	**0.785**
sink	0.616	0.645	0.581	0.618	0.676	**0.748**
sofa	0.533	0.626	0.598	0.657	0.621	0.623
table	0.375	0.365	0.429	0.450	0.470	0.284
toilet	0.820	0.748	0.760	0.788	0.762	**0.847**
wall	0.729	0.725	0.661	0.721	0.779	0.773
window	0.560	0.529	0.446	0.408	0.541	0.548
mIoU	0.475	0.485	0.498	0.503	0.535	**0.560**

## Data Availability

The raw data supporting the conclusions of this article will be made available by the authors on request.

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
