# Peer review of "DDNet: Depth Dominant Network for Semantic Segmentation of RGB-D Images"

_sensors, 2024, doi:10.3390/s24216914_

Round 1
Reviewer 1 Report
Comments and Suggestions for Authors
The paper presents a novel depth dominant network (DDNet) designed to enhance RGB-D semantic segmentation by emphasizing depth information over traditional RGB data. The proposed architecture features dual CNN branches for extracting color and depth features, with a larger fusion weight assigned to the depth branch. The study presents a significant advancement in RGB-D semantic segmentation. Addressing the following weaknesses would enhance the paper's clarity and impact, making it a strong candidate for publication.
1. What specifically are the geometric and contextual information in RGB images?
2. More visualizations and additional evidence wheather "obvious geometric information in the depth image is more conducive to segmentation than RGB data" ,"Color features and depth features were, respectively, extracted by two CNN branches." are needed.
3. The idea of using depth images as the primary input for segmentation, while utilizing RGB images with blurred edge information to provide semantic information for the depth branch as an auxiliary module, is sound. However, the authors do not seem to clearly articulate their motivation for this approach. I suggest strengthening the articulation of the motivation for the introduction, specifically explaining why a dual-branch approach is being adopted.
4. in section 3.3,The value of /alpha, /beta needs to be clearly indicated.
5. The use of a dual-branch network raises the question of whether it offers advantages in terms of parameter count and computational load compared to existing state-of-the-art methods. I suggest including relevant experiments to address this.
6. there are some mistake in paper such as in section 3.3 "where" should be in lowercase
Comments on the Quality of English LanguageN/A
Reviewer 2 Report
Comments and Suggestions for Authors
The proposed approach is novel and interesting, especially the use of DDNet for RGB sequential segmentation. The idea of ​​using tunable fusion weights for depth and RGB features is also compelling and the model's effectiveness is proven. But the following areas need improvement:
1- The compared methods are too old; most are from before 2020. It is necessary to compare it with the most recent state-of-the-art.
2- The explanation of the fusion weights is somewhat limited. More details need to be given to answer why these weights are chosen.
3- The evaluation on NYUv2 and Scan Netv2 is indeed solid, but having other datasets to evaluate the proposed model can add more value ​​on paper.
4- The network architecture and PPM could be explained more clearly. I need to see how to break down complex paragraphs and provide more descriptions about them.
5- Why do some errors occur and how does your method mitigate them? I would like to see more analysis details.
Comments on the Quality of English LanguageThe quality of the English language is generally good, but there are a few parts that could be improved in terms of clarity and readability. Additionally, there are minor grammatical errors and awkward wording in some sections.
Example:
Line 157:
It’s obvious to find that depth information is an auxiliary module of colors branch in the majority of 2D semantic segmentation.
Can be:
It is evident that depth information serves as an auxiliary module to the color branch in most 2D semantic segmentation methods.
Round 2
Reviewer 1 Report
Comments and Suggestions for Authors
The authors solved my confusions through detailed analysis, and I have no other comments.
Reviewer 2 Report
Comments and Suggestions for Authors
My concerns are resolved and the paper can be published. I thank the authors for this well-detailed paper and its edits.